# Hierarchical control of locomotion by distinct types of spinal V2a interneurons in zebrafish

Evdokia Menelaou[1,2] & David L. McLean[1]

In all vertebrates, excitatory spinal interneurons execute dynamic adjustments in the timing and amplitude of locomotor movements. Currently, it is unclear whether interneurons responsible for timing control are distinct from those involved in amplitude control. Here, we show that in larval zebrafish, molecularly, morphologically and electrophysiologically distinct types of V2a neurons exhibit complementary patterns of connectivity. Stronger higher-order connections from type I neurons to other excitatory V2a and inhibitory V0d interneurons provide timing control, while stronger last-order connections from type II neurons to motor neurons provide amplitude control. Thus, timing and amplitude are coordinated by distinct interneurons distinguished not by their occupation of hierarchically-arranged anatomical layers, but rather by differences in the reliability and probability of higher-order and last-order connections that ultimately form a single anatomical layer. These findings contribute to our understanding of the origins of timing and amplitude control in the spinal cord.

[1] Department of Neurobiology, Northwestern University, Evanston, IL 60208, USA. [2] Present address: Department of Organismal Biology and Anatomy, University of Chicago, Chicago, IL 60637, USA. Correspondence and requests for materials should be addressed to D.L.M. (email: david-mclean@northwestern.edu)

L
ocomotion is a complex task requiring the coordination of diverse pools of motor neurons and the muscles they control. To effectively navigate in unpredictable environments, all animals must differentially adjust the timing and amplitude of rhythmic oscillatory drive to different motor pools[1–3]. In vertebrates, phylogenetically conserved populations of spinal interneurons with distinct molecular and electrophysiological signatures are responsible for executing this crucial process[4–7].

To explain the dynamics of limb-based locomotor control, it has been proposed that spinal interneurons are organized as multi-layered, hierarchical arrays[8–11]. Coordination of timing signals like frequency and phase is provided by interneurons with exclusively higher-order connections to other interneurons, while adjustments in amplitude are executed by last-order interneurons that directly drive recruitment in specific motor pools (Fig. 1a, top). However, in simpler axial locomotor circuits where physiological measures of connectivity are more easily performed[10–14],

last-order excitatory interneurons are instead proposed to form a single layer that also provides higher-order connections to coordinate timing (Fig. 1a, bottom). Since navigation during swimming also requires differential control of the timing and amplitude of rhythmic activity in axial motor pools[15–18], a means to stratify locomotor control using a single layer architecture presumably exists, but has yet to be revealed. If so, this would provide better insight into the evolutionary origins of timing and amplitude control in the spinal cord.

To address this fundamental issue, we have physiologically mapped connections arising from a class of excitatory interneuron that has previously been implicated in both axial and limb locomotor control. In zebrafish, ipsilaterally projecting V2a neurons labeled by the transcription factor Chx10 provide glutamatergic and mixed electrical synaptic connections that directly excite motor neurons[19–21]. V2a neurons also provide distinct sources of excitation to epaxial and hypaxial motor pools that

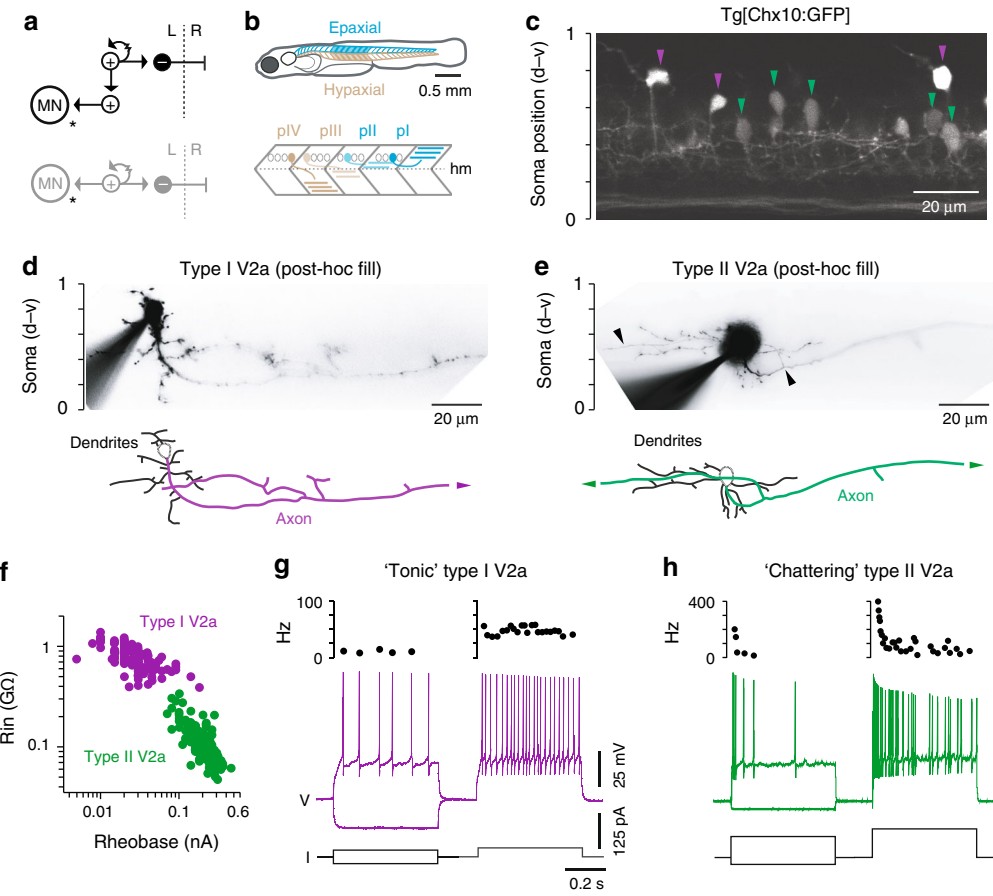

**Fig. 1** Two types of V2a neuron distinguished by intrinsic properties. **a** Schematics illustrating multi-layer (top) and single-layer (bottom) models of connectivity underlying timing and amplitude control. + excitatory interneurons, – inhibitory interneurons, MN motor neurons, L left side, R right side. Arrows are glutamatergic connections and the resistor symbol denotes electrical coupling. **b** Top, schematic illustrating a larval zebrafish viewed from the side. Region at midbody (segments 10–16) where recordings were performed is shaded. Rostral, left; dorsal, up. Bottom, schematic illustrating muscle segments and each of the four identified primary motor neurons per spinal hemisegment (dotted circles) that innervate either epaxial (pI-II) or hypaxial (pIII-IV) muscle (horizonal blue or brown lines). hm, horizontal myoseptum. **c** Single confocal optical section from one spinal segment illustrating type I (purple arrowheads) and type II (green arrowheads) V2a neurons targeted for recordings. Soma positions are normalized to the dorsal and ventral aspects of spinal cord (d-v). Rostral is to the left. **d** Top, contrast inverted epifluorescent images of fills after electrophysiological recordings. Bottom, reconstructions illustrating the trajectory of the main axon. The descending axon of the type I V2a neuron continues out of the field of view (arrowhead). Scale to the left indicates normalized dorso-ventral soma position. **e** As in panel **d**, but for a bifurcating type II V2a neuron. In top image, black arrowheads mark axon bifurcation point and the continuation of the ascending axon out of the field of view. **f** Quantification of input resistance (Rin) and rheobase according to V2a neuron type. Type I V2a (n = 105), type II V2a (n = 145). Source data are provided in a Source Data file. **g** Top, instantaneous firing frequencies are plotted above the respective recordings from a type I V2a neuron. Bottom, examples of voltage deflections in response to a hyperpolarizing current step (−25 pA) and firing responses to current steps around threshold (left trace) and just above threshold (right trace). Voltage; I, current. Source data are provided in a Source Data file. **h**, As in panel **g**, but for a type II V2a neuron

allow for rapid adjustments in posture (Fig. 1b)[22]. This drive could originate from two sources, V2a neurons with descending projections within spinal cord and V2a neurons with descending and supraspinal projections, both of which appear to make perisomatic last-order connections[23].

In mice, V2a neurons are also divided into two distinct types (type I and II) based on intraspinal versus supraspinal projection patterns and levels of Chx10 expression[24]. Molecular and anatomical studies have demonstrated that both types of V2a neurons are glutamatergic, can form electrical synapses and make last-order connections[24–28]. Physiological studies also suggest that V2a neurons can be distinguished by differences in their electrical coupling, firing properties and their participation in fictive locomotor-related activity[29–31]. While this level of V2a diversity is consistent with the potential for distinct roles during locomotion, the fact that both types of V2a neuron contact motor neurons is difficult to reconcile with a multi-layer framework.

Here, we reveal differences in the molecular, morphological, and electrophysiological properties of V2a neurons in larval zebrafish that mirror those reported in mice. We find that distinct types of V2a neurons are not organized as anatomically distinct higher-order and last-order multi-layer arrays that stratify timing and amplitude control. Instead, distinct types of V2a neurons exhibit differences in the relative modes, weights and probabilities of higher-order and last-order connections that functionally stratify control, but operate as a single, interconnected layer. By providing the first physiological assessment of connectivity between molecularly defined spinal interneurons and motor neurons, our findings reconcile observations in both axial and limb circuits and help explain the origins of locomotor coordination in the spinal cord.

## Results

**Distinct intrinsic properties of spinal V2a neurons**. In zebrafish, different spinal V2a neurons are engaged at different speeds of locomotion[21,32]. Thus, to simplify our assessment of connectivity using paired patch-clamp recordings, we focused on V2a neurons exclusively engaged at high frequencies of swimming, which are less numerous, easily identifiable and occupy dorsal locations in the spinal cord[23,33]. We first assessed whether morphologically distinct V2a neurons could also be distinguished by levels of Chx10 expression and electrophysiological properties, as reported in mice[24,29,30].

In Tg[Chx10:GFP] larval fish[19], we observed relatively brightly labeled somata displaced dorsally from the main V2a population and dimly labeled somata located just below them (Fig. 1c). Post-hoc fills following electrophysiological recordings revealed that brighter GFP-labeled V2a neurons were descending (Fig. 1d), while dimmer GFP-labeled V2a neurons were bifurcating (Fig. 1e). A comparison of the maximum intensity of bright and dim dorsal V2a neurons in confocal images revealed an almost two-fold difference in brightness (means ± standard deviations, 244 ± 11 a.u. versus 145 ± 26 a.u., $n = 9$; Student's $t$-test, t(16) = 10.3, $p < 0.001$). This matches observations in mice, so from here on we will refer to brighter, descending V2as as type I and dimmer, bifurcating V2as as type II.

The distinct molecular and morphological features characterizing type I and II V2a neurons were also accompanied by distinct electrophysiological properties. Type I V2a neurons had higher input resistances and lower rheobase values (Fig. 1f), consistent with differences in soma size between the types (type I: 25 ± 6 μm², $n = 105$; type II: 35 ± 7 μm², $n = 145$; Mann–Whitney $U$-test, U(248) = 1690, $p < 0.001$). In response to direct current injection just above threshold, type I V2a neurons generated a stable tonic firing response, characterized by spike frequencies ranging from 10–250 Hz (72 ± 50 Hz, $n = 96$), which were sensitive to increasing levels of current, but exhibited relatively little accommodation (Fig. 1g). On the other hand, type II neurons exhibited a more accommodative chattering firing pattern, with a barrage of two or more high frequency spikes at the beginning of the step that continued intermittently throughout (Fig. 1h). Instantaneous spike frequencies just above threshold were also current sensitive, but ranged from 130–650 Hz (372 ± 99 Hz, $n = 140$).

Differences in size and firing rates between types prompted us to assess whether there were also differences in conduction velocity (Fig. 2a). Simultaneous somatic and axonal recordings revealed that somatic spike heights did not differ significantly (Student's $t$-test, t(9) = 2.3, $p = 0.168$). However, the resulting axonal spikes were significantly smaller (Student's $t$-test, t(9) = 2.3, $p < 0.05$) and conducted more slowly in type I V2as (Student's $t$-test, t(9) = 2.3, $p < 0.001$). Specifically, spikes propagated around 0.5 m/s in type II V2as (0.48 ± 0.04 m/s, $n = 4$), compared 0.2 m/s in type I V2as (0.22 ± 0.03 m/s, $n = 7$), meaning type II V2a neurons can convey information more quickly.

However, despite these differences in intrinsic properties both types of V2a neuron were preferentially recruited at high-frequencies of swimming. In response to brief cutaneous electrical stimulation, episodes of fictive swimming can be evoked and the frequency of local motor bursts monitored by peripheral motor nerve recordings (Fig. 2b). Current-clamp recordings from both V2a types revealed oscillations in membrane potential coincident with cyclical motor bursts, with spiking observed only during the highest frequencies immediately following the stimulus (Fig. 2b). Quantification of the reliability of firing as a function of swim frequency revealed both types were more reliably active at higher frequencies, reaching 100% above 60 Hz (Fig. 2c). Notably, this frequency range overlaps with the intrinsic firing frequencies of type I V2a neurons.

If V2a neurons are responsible for generating high-frequency swimming rhythms, we would expect their activity to consistently lead the onset of local motor activity. To test this, we performed recordings from V2a neurons and primary motor neurons located in the same spinal segment (Fig. 2d). Primary motor neurons are large identified neurons that also participate exclusively in high-frequency swimming and are divided into distinct epaxial and hypaxial pools (Fig. 1b)[34]. In both type I and II V2a neurons spikes coincided with cyclical primary motor neuron activity (Fig. 2d). When we quantified the distribution of V2a neuron spikes relative to the first spike in primary motor neurons, spiking in both types preceded the onset of primary motor neuron activation by ~5 ms and trailed it by ~10 ms (Fig. 2e, f). For type II V2a neurons spiking peaked at the onset of the first spike (Fig. 2e), while for type I V2a neurons the peak in spike distribution was slightly delayed (Fig. 2f). This subtle difference is likely explained by differences in membrane time constants shaping responses to coincident rhythmic excitatory drive.

Thus, V2a neurons that participate exclusively in higher frequencies of swimming in larval zebrafish also can be divided into two molecularly, morphologically, and electrophysiologically distinct types. In addition, both V2a neuron types fire cyclically prior to and during motor output, consistent with a role in generating and sustaining motor bursts during high-frequency rhythms.

**Different patterns of interconnectivity among V2a neurons**. In mice, both types of V2a neurons express connexin-36[24] and electrical coupling between distinct types of V2a neurons is reportedly stronger within types than between them[30]. Glutamatergic interconnections have not been confirmed physiologically, although

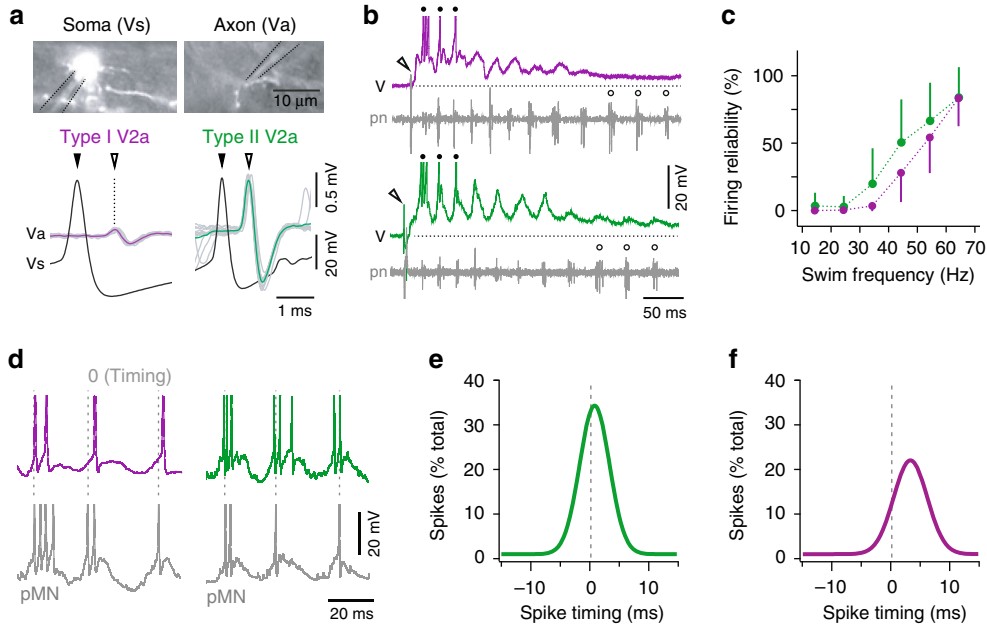

**Fig. 2** Distinct V2a neuron types are recruited at higher swimming frequencies. **a** Top, images of type I V2a neuron mosaically labeled with green fluorescent protein to obtain simultaneous intracellular somatic (Vs) and extracellular axonal (Va) recordings for conduction velocity measurements. Dashed lines indicate recording electrodes. Bottom, intracellular somatic and extracellular axonal recordings from type I and II V2a neurons. Somatic spikes are presented as mean waveform (filled arrowheads), while for axonal spikes the mean (open arrowheads) is superimposed on individual traces (gray). **b** Top, simultaneous current-clamp recordings from a type I V2a neuron (V) and extracellular recordings of peripheral motor nerves (pn) during fictive swimming following a brief electrical stimulus to the skin (open arrowhead). Filled circles highlight suprathreshold swimming frequencies, while open circles highlight examples of subthreshold frequencies. Spikes are truncated to better reveal the underlying synaptic drive. Dashed line indicates resting membrane potential, −53 mV. Bottom, as above but for a type II V2a neuron. Resting membrane potential, −70 mV. **c** Quantification of the reliability of firing as a function of swimming frequency for type I (n = 21) and type II (n = 55) V2a neurons. Data are reported as means ± standard deviations. Source data are provided in a Source Data file. **d** Current clamp recordings of excerpts of suprathreshold rhythmic activity during fictive swimming recorded from type I (left) and type II (right) V2a neurons and a primary motor neuron (pMN) in the same spinal segment. Spikes are truncated to better reveal the underlying synaptic drive. Gray dashed lines indicate the first spike in the pMN used to calculate V2a spike timing. **e** A gaussian fit of spike timing relative to the first spike in a local primary motor neuron (0, dashed line) for type II V2a neurons (n = 7). See Methods for details. Source data are provided in a Source Data file. **f** As in panel **e**, but for type I V2a neurons (n = 7)

there is anatomical evidence for glutamatergic V2a–V2a connections that have yet to be linked to type[24]. To see how these observations hold for zebrafish, we next assessed connections within and between V2a types using paired patch-clamp recordings.

In recordings within type I V2a neurons (n = 12), we evoked excitatory post-synaptic potentials (EPSPs) and post-synaptic currents (EPSCs) that were unidirectional, consistent with their descending morphologies (Fig. 3a, left). EPSPs between type I V2as were either electrical or mixed and peak amplitudes could reach 2 mV (Fig. 3c). Electrical synapses were characterized by the absence of failures, relatively fixed amplitudes (Fig. 4a) and their sensitivity to the gap junction blocker, 18βGA[35], but not the AMPA-receptor antagonist NBQX (Fig. 4b, c). Mixed synapses were characterized by dual component EPSP/EPSCs (Fig. 4a), with shorter latency electrical components sensitive to 18βGA and longer latency components with variable amplitudes and failures that were largely sensitive to NBQX, consistent with a glutamatergic connection (Fig. 4b, c).

In recordings within type II V2a neurons (n = 15), we observed slower, lower amplitude interactions, which were bidirectional consistent with their bifurcating morphologies (Fig. 3a, right). On a compressed time scale, slow responses resembled EPSPs and EPSCs (Fig. 4a, inset) and were most sensitive to 18βGA application (Fig. 4b, c), consistent with an electrical connection. Slow responses were easily distinguishable from other electrical synapses based on their kinetics (Fig. 4d). While the majority of interactions between type II V2a neurons were slow (Fig. 3c), in

two recordings we also observed relatively weak electrical synapses with faster kinetics that preceded slow responses (Fig. 3a, inset), but were only observed in the descending direction.

When we performed recordings between types we also observed descending synaptic interactions (Fig. 3b). However, in contrast to connections within types that were primarily electrical, connections between types were primarily chemical in nature (Fig. 3d). Chemical connections exhibited the same characteristics and pharmacological sensitivity as the glutamatergic component of mixed synapses (Fig. 4a–c). In the ascending direction between types, from caudal type II V2a neurons to rostral type I V2a neurons we consistently observed slow responses (Fig. 3b, left). However, caudal type I V2a neuron interactions to rostral type II V2a neurons were never observed (Fig. 3b, right).

Critically, when we compared the peak amplitudes of descending glutamatergic synaptic connections, there was a clear functional asymmetry between the two V2a neuron types. EPSPs from rostral type I V2a neurons to caudal type II V2as (n = 7) were relatively strong and could reach peak amplitudes of 5 mV (Fig. 3d). On the other hand, EPSPs from rostral type II V2a neurons to caudal type I V2a neurons (n = 5) rarely exceeded 1 mV and were significantly smaller on average (Fig. 3d).

Thus, connections within distinct types of V2a neuron are primarily electrical, however connections between types include descending glutamatergic connections that are functionally asymmetric. Specifically, inputs from type I V2a neurons are

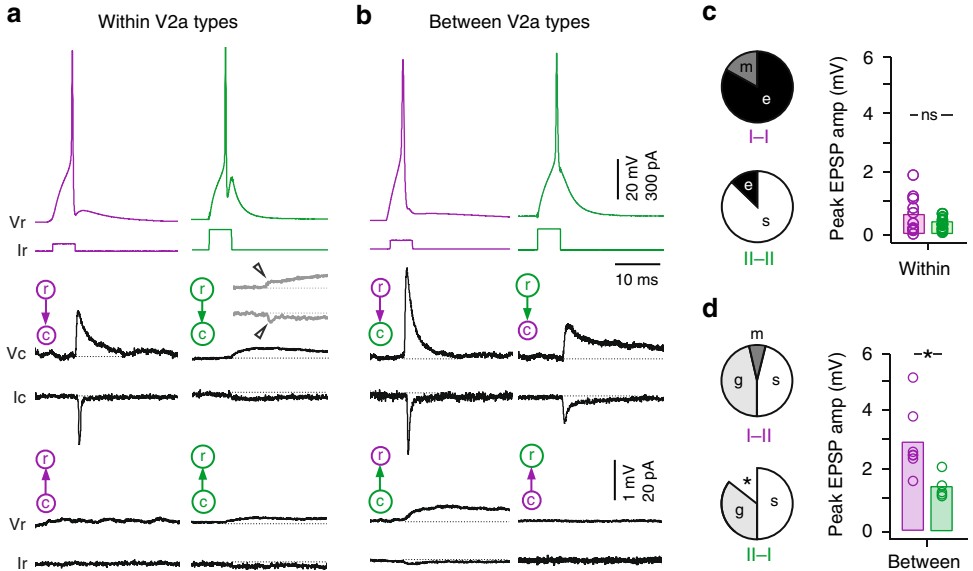

**Fig. 3** Distinct patterns of connectivity within and between V2a types. **a** Left, example of paired recording within type I V2a neurons (purple) testing connectivity in the descending (top and middle traces) and ascending directions (bottom traces), as indicated by cartoons (r rostral, c caudal). Right, example of a paired recording within type II V2a neurons (green). For simplicity, spikes evoked in caudal neurons are not illustrated. Spikes are individual sweeps, while post-synaptic responses are spike triggered averages of all events. I, current; V, voltage. Inset gray trace is from another paired type II V2a neuron recording on the same scale, illustrating the weak descending electrical synapse (at arrowheads). **b** As in panel **a**, but paired recordings between type I and II V2a neurons. Middle traces are averages of successful events. **c** Left, quantification of the mode of descending synaptic connectivity within different types of V2a neurons expressed as a percentage of total recordings (I to I, $n = 12$; II to II, $n = 15$). e electric, m mixed, s slow. Right, quantification of the peak amplitudes of EPSPs (electric, mixed, glutamatergic or slow) measured within type I (purple, $n = 12$) and type II (green, $n = 15$) V2a neurons. ns, not significant following Mann–Whitney U-test, U(25) = 105, $p = 0.464$. Bars represent means. Source data are provided in a Source Data file. **d** Left, quantification of the mode of descending synaptic connectivity between types expressed as a percentage of total recordings (I to II, $n = 13$; II to I, $n = 14$). Asterisk marks proportion of total recordings where no connection was observed ($n = 2$). g glutamatergic. Right, quantification of the peak amplitudes of glutamatergic EPSPs measured from type I to II V2as (purple, $n = 7$) and from type II to I V2as (green, $n = 5$). * significant difference following Mann–Whitney U-test, U(10) = 34, $p < 0.01$. Source data are provided in a Source Data file

stronger to type II V2a neurons. This suggests that type II V2a neurons ultimately rely more on input from type I V2as to shape their rhythmic activity than vice versa.

**Different patterns of V2a connectivity with V0d neurons.** Since type I V2a neurons in zebrafish have stronger high-order excitatory interneuron connections, we next assessed whether this would also extend to connections with inhibitory interneurons. In mice, it has been proposed that type I V2a neurons innervate commissural interneurons to control the timing of left–right alternation during locomotion[31]. One potential target based on anatomical evidence is the inhibitory V0d population labeled by the transcription factor Dbx1[36–39]. In larval zebrafish, glycinergic dbx1-positive V0d neurons are morphologically indistinguishable from a class of commissural bifurcating longitudinal (CoBL) interneuron implicated in left–right alternation during swimming[33,40,41].

To target V0d-CoBL neurons for recordings we used compound Tg[GlyT2:lRl-GFP × Dbx1:Cre] fish (Fig. 5a)[41] or Tg[GlyT2:GFP] fish[33]. Inhibitory interneurons are also recruited differentially according to speed[33]. Consequently, we focused on the most ventral V0d neurons recruited at higher frequencies that would likely be targeted by fast V2a neurons (Fig. 5a). By focusing on the ventral-most cells in the Tg[GlyT2:GFP] line, this also reduced the possibility of recording from CoBL interneurons derived from more dorsal progenitor domains[42]. Post-hoc fills using either approach revealed neurons with indistinguishable morphologies and recruitment patterns, namely a spherical soma

with short dendritic processes from the ipsilateral axon and collaterals that project dorsally from the commissural bifurcating axon (Fig. 5b). In current-clamp recordings during fictive swimming, ventral V0d neurons also fired rhythmically immediately following a brief cutaneous stimulus (Fig. 5c) and were recruited preferentially at higher swim frequencies (Fig. 5d).

When we assessed V2a to V0d connectivity in the descending direction, we found that both types of V2a neurons formed connections with V0d neurons. Type I V2a inputs to V0ds ($n = 11$) were either electrical, mixed or glutamatergic (Fig. 5e). Type II V2a inputs to V0d neurons ($n = 17$) were either electrical, mixed or slow (Fig. 5e). Critically, however, even when excluding lower amplitude, slow type II V2a to V0d connections, type I V2a to V0d connections were significantly larger in amplitude (Fig. 5e).

Thus, both types of V2a neuron form glutamatergic and electrical synapses with V0d neurons. However, type I V2as provide a stronger source of higher-order excitation and likely play a more dominant role in coordinating V0d activity, as they do for other V2a neurons.

**Distinct patterns of last-order V2a connectivity.** Anatomical and viral tracing methods in mice suggest that both types of V2a neuron form last-order connections[24–26]. Our results thus far suggest a more nuanced circuit architecture, where different types of interconnected V2a neurons exhibit differences in the strength of higher-order connections. To see whether type-specific differences extended to last-order connections, we performed paired

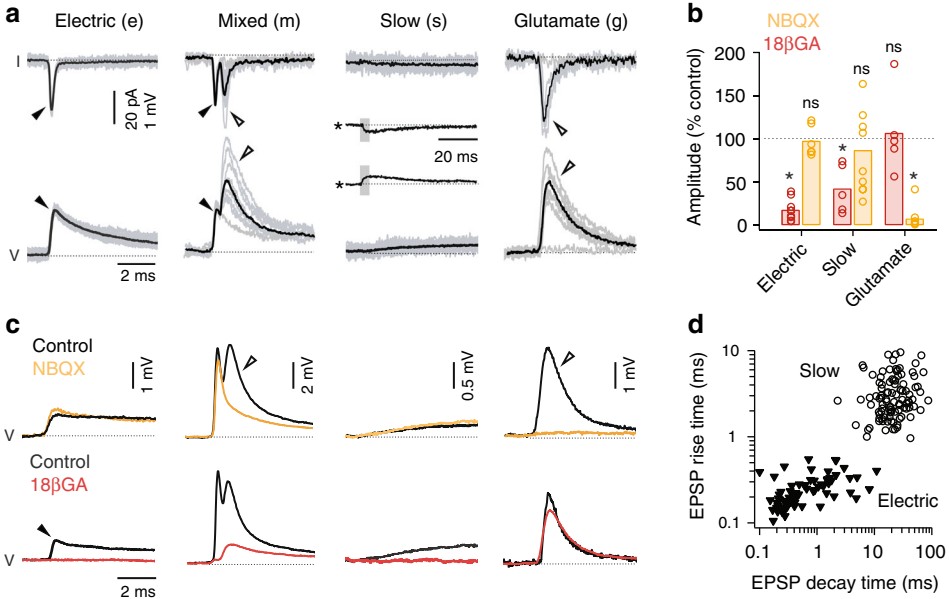

Fig. 4 V2a neurons exhibit diverse modes of synaptic connectivity. **a** Recordings of spike-triggered post-synaptic events in voltage-clamp (top row) and current-clamp (bottom row). Filled arrowheads mark electrical excitatory post-synaptic currents (EPSCs) and post-synaptic potentials (EPSPs), while open arrowheads mark glutamatergic EPSCs and EPSPs. Slow electrical EPSCs and EPSPs are also inset on a compressed time scale (asterisks), with a gray box indicating the expanded region above or below. Black lines are averages of successful events and gray lines are individual events, including failures where appropriate. I current, V voltage. **b** Quantification of the effect of drug treatment normalized to control EPSP or EPSC amplitudes for electric, glutamatergic (glut.) and slow responses. Electric 18βGA (* significant following Student's t-test, t(16) = 19.9, p < 0.001, n = 9); electric NBQX (ns, not significant following Student's t-test, t(10) = 0.4, p = 0.711, n = 6); glutamatergic 18βGA (ns, Student's t-test, t(8) = 0.3, p = 0.772, n = 5); glutamatergic NBQX (*Student's t-test, t(32) = 38.9, p < 0.001, n = 17); slow 18βGA (* Student's t-test, t(8) = 4.6, p < 0.01, n = 5); slow NBQX (ns, Student's t-test, t(24) = 1.1, p = 0.263, n = 13). Bars represent means. Source data are provided in a Source Data file. **c** Spike triggered averages of successful post-synaptic responses recorded in current clamp before and after drug treatment. Data are organized from left to right according to type of response, as in panel **a**. **d** Quantification of EPSP rise versus decay time for all electric (n = 77) and slow responses (n = 101). Electric EPSPs are easily distinguishable from slow responses based on time course. Source data are provided in a Source Data file

patch-clamp recordings between type I (n = 38) or type II V2a neurons (n = 71) and identified pairs of epaxial and hypaxial primary motor neurons (Fig. 1b).

Both types of V2a neurons formed synaptic connections with epaxial and hypaxial motor neurons, consistent with their role in generating and sustaining motor output. However, last-order inputs from type I V2a neurons were glutamatergic or mixed, while inputs from type II V2a neurons were either mixed, electrical or slow (Fig. 6a). Notably, the failure rate for all purely glutamatergic synapses was significantly higher than the glutamatergic component of mixed synapses (glutamatergic, 80 ± 18%, n = 47 versus mixed, 65 ± 27%, n = 50; Mann–Whitney U-test, U (95) = 1568, p < 0.01). Thus, last-order inputs from type II V2a neurons were more reliable. In addition, peak amplitudes of mixed or electric EPSPs from type II V2a neurons were also significantly larger than mixed or glutamatergic EPSPs from type I V2a neurons (Fig. 6a).

Since type I V2a neurons always formed fast last-order connections, albeit with high failure rates, this left type II V2a neurons as the likely source of the fast epaxial and hypaxial pool-specific input reported previously for self-righting maneuvers in larval zebrafish[22]. To directly test this idea, we performed dual re-patch experiments where epaxial and hypaxial motor neurons were sequentially sampled while holding the same presynaptic V2a neuron (Fig. 6b). For type I V2a to motor neuron recordings fast synaptic connections were observed in both epaxial and hypaxial sequential recordings (Fig. 6c, left). In contrast, for type II V2a neurons when fast responses were observed in either the epaxial or hypaxial motor neuron, only a slow response was observed in the other (Fig. 6c, right).

Given the distinctive nature of slow responses and the preponderance of electrical synaptic connections, we were concerned that slow responses may in fact represent indirect electrical continuity, either via common target neurons[43] and/or a shared source of reticulospinal input[44]. To test the latter possibility more directly, we measured electrical coupling coefficients between primary motor neurons separated by 4 body segments, where there is no chance of a direct physical contact (Fig. 7a). In support of indirect electrical continuity, we observed bidirectional propagation of hyperpolarizing current steps between distal motor neurons (Fig. 7b, c). When we performed a similar analysis between different types of V2a neurons separated by 2–4 body segments, where physical contact via presynaptic axons is the only possibility, a similar level of coupling was observed (Fig. 7b, d, e). Remarkably, however, when assessing V2a neurons and all of their post-synaptic targets, electrical coupling was observed in the caudo-rostral direction regardless of the mode of synaptic transmission, whether slow, electrical, glutamatergic or mixed (Fig. 7f). Collectively, these data suggest that electrical coupling recorded between distal neurons does not report the existence of fast interconnections and that slow responses are more likely to reflect indirect electrical interactions.

The fact that a 50/50 likelihood of direct, fast responses versus indirect slow ones reflects pool-specific inputs prompted us to reconsider the probability of finding fast higher-order V2a connections. When considering only fast electrical, mixed or glutamatergic connections, type I V2a neurons had a connection probability of 1 (Fig. 7g). The one exception was connections to type II V2a neurons, which were around 0.5 (Fig. 7g). The

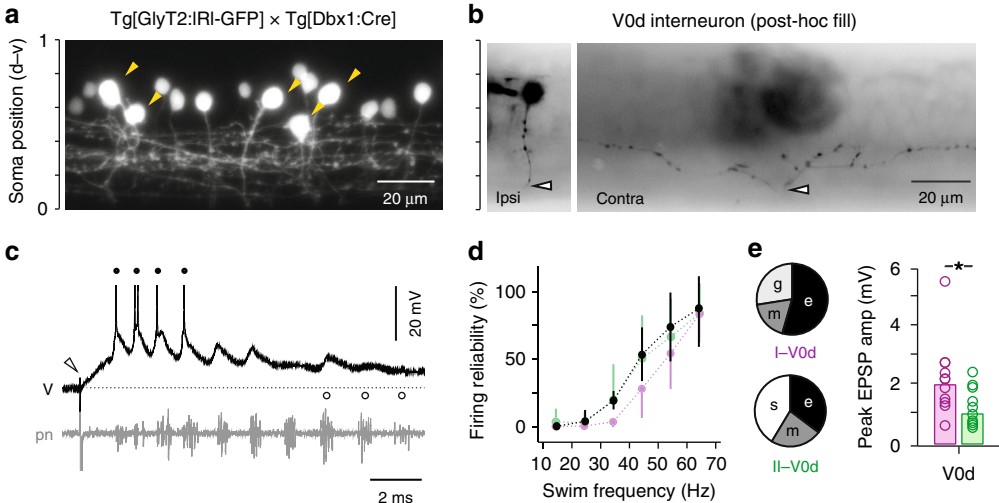

**Fig. 5** V0d inhibitory interneuron recruitment and targeting by V2a types. **a** Single confocal optical section from one spinal segment in compound transgenic larvae illustrating the location of V0ds targeted for recordings (yellow arrowheads). Note, only GFP fluorescence is illustrated. Scale to the left indicates normalized dorso-ventral soma position. **b** Contrast inverted epifluorescent images after recording illustrating the ipsilateral soma (ipsi) and contralateral axon (contra). The axon crosses at the white arrowheads. Soma position is normalized to dorso-ventral aspects of spinal cord, as in panel **a**. **c** Simultaneous current-clamp recordings from a V0d neuron (V) and extracellular recordings of peripheral motor nerves (pn) during fictive swimming following a brief electrical stimulus to the skin (open arrowhead). Filled circles highlight suprathreshold swimming frequencies, while open circles highlight examples of subthreshold frequencies. Spikes are truncated to better reveal the underlying synaptic drive. Dashed line indicates resting membrane potential, −54 mV. **d** Quantification of the reliability of firing as a function of swimming frequency for V0d neurons ($n = 26$). Values are superimposed on V2a data from Fig. 2c. Data are reported as means ± standard deviations. Source data are provided in a Source Data file. **e** Left, quantification of the mode of V2a synaptic connectivity to V0d neurons expressed as a percentage of total responses (I to V0d, $n = 11$; II to V0d, $n = 17$). e electric, m mixed, g glutamatergic, s slow. Right, quantification of the peak amplitudes of fast EPSPs (electric, mixed, or glutamatergic) measured from type I V2a to V0d neurons (purple, $n = 11$) and type II V2a to V0d neurons (green, $n = 10$). * significant difference following Mann–Whitney $U$-test, U(19) = 84, $p < 0.05$. Bars represent means. Source data are provided in a Source Data file

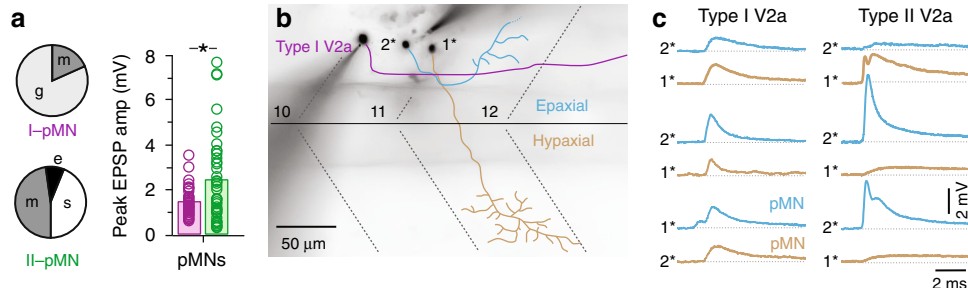

**Fig. 6** Distinct patterns of last-order V2a connectivity. **a** Left, quantification of the mode of V2a synaptic connectivity to primary motor neurons (pMN) expressed as a percentage of total responses (I to pMN, $n = 38$; II to pMN, $n = 71$). e electric, m mixed, g glutamatergic, s slow. Right, quantification of the peak amplitudes of fast EPSPs (electric, mixed, and glutamatergic) measured from type I V2a to primary motor neurons (purple, $n = 38$) and type II V2a to primary motor neurons (green, $n = 40$). * significant difference following Mann–Whitney $U$-test, U(76) = 580, $p < 0.05$. Bars represent means. Source data are provided in a Source Data file. **b** Contrast inverted fluorescent images of fills post-recording from a type I V2a neuron (purple) and an epaxial (blue) and hypaxial (brown) primary motor neuron. 1*, targeted first; 2*, targeted second. Segment boundaries and the horizontal myoseptum are marked by black dashed lines. Body segments are numbered. **c** Average excitatory post-synaptic potentials recorded from hypaxial primary motor neurons (brown) and epaxial motor neurons (blue) for type I V2a neurons (left) and type II V2a neurons (right). Each row represents a different experiment ($n = 3$ for each). Order of post-synaptic recording is indicated by 1* (first) and 2* (second)

probability of type II V2a higher-order connections was consistently around 0.5, excluding sparse descending connections to other type II V2as (Fig. 7g). Notably, if the lack of a fast connection in the ascending direction was also considered, the probability of type II V2a interconnections would be even lower.

Thus, type II V2a neurons form stronger, more reliable last-order connections that provide drive to either epaxial or hypaxial motor pools, while type I V2a neurons form weaker, less reliable last-order connections that provide drive to both epaxial and hypaxial pools. These last-order patterns complement the higher-order connections originating from distinct V2a neuron types.

Consequently, it is conceivable that the pool-specific nature of last-order inputs is also preserved at the level of higher-order connections between type I and II V2a neurons and from type II V2a to V0d neurons.

## Discussion

Our goal was to see how spinal interneurons are synaptically interconnected to explain the basis of timing and amplitude control during locomotion. To do so, we took advantage of the identifiable nature and limited number of phylogenetically

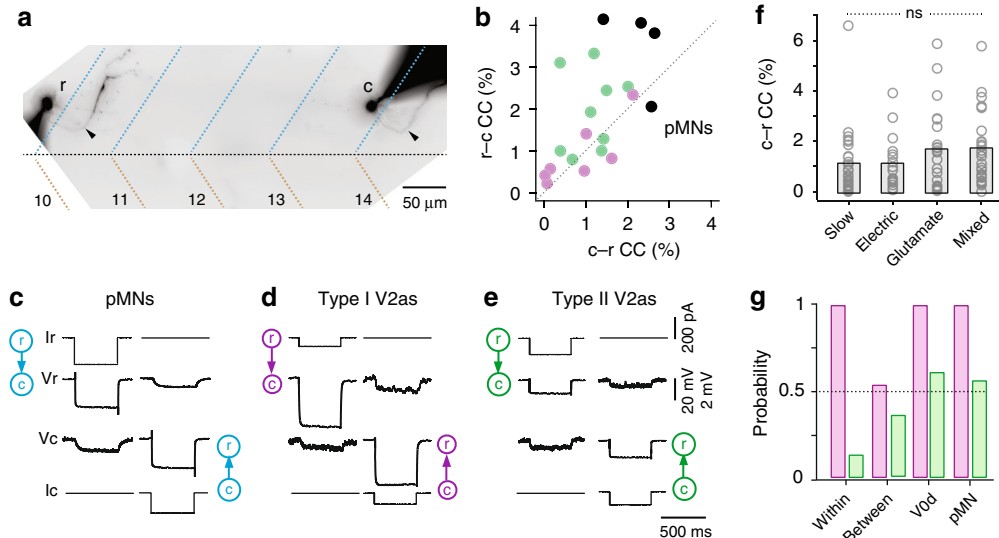

**Fig. 7** Electrical coupling unrelated to fast synaptic connectivity. **a** Contrast inverted fluorescent images of fills post-recording from rostral and caudal epaxial primary motor neurons separated by four body segments, whose coupling is illustrated in panel c. Filled arrowheads indicate axon innervating the dorsal, epaxial muscle (blue dashed lines), which is separated from ventral, hypaxial muscle (brown dashed lines) by the horizonal myoseptum (black dashed line). Muscle segments are numbered. **b** Quantification of coupling coefficients (CC) within type I V2as (purple, $n = 7$ pairs), type II V2as (green, $n = 9$ pairs) and primary motor neurons (black; $n = 2$ epaxial/epaxial, $n = 1$ hypaxial/hypaxial, $n = 1$ epaxial/hypaxial) in both descending (rostro-caudal, r-c) and ascending (c-r) directions. All recordings were performed between neurons separated by 2–4 body segments. Source data are provided in a Source Data file. **c** Paired recordings from epaxial motor neurons in panel a illustrate coupling revealed by hyperpolarizing current steps in both descending (r-c) and ascending (c-r) directions, as indicated by cartoons. V voltage, I current. **d** As in panel **c**, but coupling between type I V2a neurons. **e** As in panel **c**, but coupling between type II V2a neurons. **f** Quantification of the coupling coefficient measured in the ascending direction (caudo-rostral) for V2a connections to all post-synaptic targets that are slow, electric, glutamatergic (glut.) or mixed. ns not significant following Kruskal Wallis ANOVA for comparisons across multiple samples ($H(3) = 5.6$, $p = 0.131$, $n = 104$). Bars represent means. Source data are provided in a Source Data file. **g** Quantification of the probability of finding a connection from type I V2a neurons (purple) and type II V2a neurons (green) within types (type I V2a, 12/12; type II V2a, 2/15), between types (type I to type II V2a, 7/13; type II to type I V2a, 5/14), to V0d neurons (type I V2a, 11/11; type II V2a, 10/17) and to primary motor neurons (type I V2a, 38/38; type II V2a, 40/71)

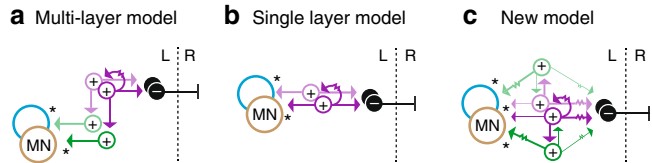

**Fig. 8** Circuit models to explain adjustments in timing and amplitude. **a** Predicted connectivity patterns for type I (purple) and II (green) V2a neurons in a multi-layer model for differential control of the timing and amplitude of activity in epaxial (blue) and hypaxial (brown) motor pools. Arrows are glutamatergic connections and the resistor symbol denotes electrical coupling. L left side, R right side. **b** as in panel **a**, but the predicted connectivity patterns of V2a neurons according to single layer models. **c** as in panel **a**, but the connectivity pattern revealed in this study in zebrafish. Larger arrows/lines indicate stronger connections

conserved spinal neurons in larval zebrafish, combined with electrophysiological approaches that enabled assessments of synaptic dynamics. According to multi-layer models, the prediction was that distinct V2a neurons would occupy separate anatomical layers dictated by higher-order versus last-order connections. According to single layer models, V2a neurons should exhibit both higher-order and last-order connections. Our data support aspects of both models. Zebrafish V2a neurons are molecularly, morphologically and electrically heterogeneous, as predicted by multi-layer models, however both types form higher-order and last-order connections, as predicted by single layer models. Instead, the functional distribution of timing and amplitude control is achieved by complementary patterns of higher-order and last-order connectivity, with stronger connections from type I neurons to other excitatory V2a and inhibitory V0d interneurons for frequency and phase control and stronger connections from type II neurons to motor neurons for amplitude control.

Hierarchical models arose to explain observations during limbed locomotion where timing and amplitude can be regulated independently[45]. Adjustments during locomotion can be defined as resetting, where perturbations lead to adjustments in the frequency and phase of on-going locomotion, or non-resetting, where perturbations do not impact the frequency and phase. The idea is that resetting inputs target type I V2a neurons with exclusively higher-order connections, while non-resetting adjustments target type II V2a neurons with exclusively last-order connections (Fig. 8a). Independent adjustments in timing and amplitude are more challenging to explain if the same V2a interneurons are responsible for both (Fig. 8b). While distinct types of V2a neurons suggest functional diversity consistent with hierarchical models, viral tracing data suggest that both types of V2a neurons contact motor neurons[24]. So how do we explain flexibility in timing and amplitude with a single layer architecture? Our data now provide a solution to this problem (Fig. 8c).

Here we show that resetting adjustments are most likely provided by type I V2a neurons, which fulfill many of the criteria predicted in multi-layer and single-layer models for interneurons controlling timing. Type I V2a neurons are descending, which ensure the rostro-caudal propagation of rhythmic activity within spinal cord during locomotion[46–48]. They have dense electric and sparse glutamatergic interconnections[49,50] and intrinsic firing

rates that could help set the frequency of locomotion, either as pacemakers or as conditional oscillators dependent on reciprocal inhibition[51–54]. V2a neurons classified here as type I are targeted directly by primary afferents in zebrafish, which is another prediction for interneurons controlling timing[55]. Type I neurons also provide strong glutamatergic drive to type II V2a neurons to control movement amplitude and strong mixed electrical/glutamatergic drive to V0d neurons to ensure left-right alternation at higher frequencies of swimming.

Non-resetting adjustments can be provided by type II V2a neurons, which fulfill many of the criteria for interneurons controlling amplitude, something predicted only in multi-layer models. They exhibit higher frequency, accommodating intrinsic firing rates better suited for controlling the intensity or duration of motor bursts[56–59]. They also lack synaptic interactions and provide strong mixed glutamatergic/electric inputs to either epaxial or hypaxial motor neurons, which allows them to act as independent relay channels. The faster conduction velocities of type II V2as likely enable motor neurons to spike earlier and more often, as required for control of movement amplitude. Anatomical studies of V2a neurons now classified as type II suggest they receive direct input from large reticulospinal neurons[60] and play a key role in regulating the amplitude of turning movements[20]. Type II V2a neurons also excite V0d neurons via mixed synapses, suggesting that pool-specific adjustments in amplitude can be transmitted quickly and reliably to contralateral epaxial/hypaxial motor pools as well[61].

Critically, because type I V2a neurons also excite motor neurons and type II V2a neurons also excite interneurons, albeit more weakly, the functional stratification we reveal here is not strictly anatomical (Fig. 8c). This explains why viral-tracing methods in mice label both types of V2a neurons and supports the topological assertions of single layer models. In fact, the interleaved nature of timing and amplitude control could explain the difficulty in disentangling frequency, phase and amplitude control using deletion strategies in mice[49]. The operational redundancy provided by distinct interneurons with differences in their relative mode, weight and probability of synaptic transmission to common targets would be difficult to tease apart, even with targeted ablations. The target-specific differences in the modes of transmission revealed here suggest that type I V2a can exert more reliable timing control via electrical synapses to other type I V2as and V0ds, but less reliable amplitude control via glutamatergic synapses to type II V2as and motor neurons. Similarly, electrical synapses from type II V2a neurons to motor neurons and V0ds ensure amplitude control will be more reliable than glutamatergic synapses to type I V2a neurons for timing control.

The ability of our new model to explain locomotor control in larval zebrafish can be summarized as follows. An abrupt stimulus would lead to the activation of type II V2a neurons at shorter latencies to provide an initial postural adjustment. Rhythmic swimming movements would then be performed via recruitment of type I V2a neurons. During swimming, type I V2a neurons provide a pool-generic envelope of rhythmic excitatory drive superimposed upon pool-specific adjustments in amplitude carried by type II V2a neurons. The relatively weak higher-order connections from type II V2as to type I V2as and V0ds would limit their impact on timing, however resetting adjustments could also be achieved if excitatory drive is sufficiently high to type II V2as. In this scenario, the ascending branch of type II V2a neurons likely serves to relay an internal copy of integrated locomotor commands to brainstem circuitry to gate on-going afferent and/or re-afferent signals[62].

While the interleaved connections between types would ensure amplitude adjustments via type II V2a neurons are appropriately coordinated with propulsive movements, for example when diving, surfacing or turning during swimming[63–65], the relative lack of connections between type II V2a neurons could also explain how zebrafish generate discrete bilateral dorsal flexions for prey strikes[66,67], torsional flexions for self-righting responses[22] and unilateral flexions for orienting maneuvers[20,68,69]. Thus, a feed-forward connectivity scheme with parallel type II circuits embedded within a serial type I circuit enables both the segregated and integrated control of rhythmic and discrete movements (Fig. 8c).

So what are the implications for spinal motor control beyond zebrafish? The relative participation of spinal interneurons in rhythmic versus discrete movements is still debated[70,71]. Recent work in mice suggests that type II V2a neurons play a key role in executing reach-and-grasp movements[25], which would be consistent with their role in discrete movements proposed here. While type classifications in mice rely on either molecular, morphological or physiological criteria, our work suggests that these features also reflect differences in connectivity and function. Consequently, one prediction is that distinct last-order primitives formed by synaptically independent type II V2a neurons can be co-opted by type I V2a neurons during locomotion or by descending commands during reach-and-grasp to execute similar movement sequences[72]. Also, since in mice both types of V2a neurons form connections with motor neurons based on viral labeling methods[24], we would predict that there would be differences in the mode, weight and probability of those connections based on type. Another prediction is that inhibitory interneurons, including V0d neurons, may also be organized into distinct types with stronger higher-order versus last-order connections to help coordinate timing and amplitude signals across the body. The distributed, but interleaved network topology we reveal here for timing and amplitude control in spinal cord could also extend to brainstem and cortical systems, dividing them into different types based on complementary connections to similar targets. Future computational work implementing the connectivity schemes revealed here will help further evaluate the functional implications for spinal motor control.

If the segregation of V2a neurons into distinct types for timing and amplitude control is a general feature of motor circuit organization, then how far might it go back? In animals closer to the base of vertebrate lineage, the anatomical and synaptic heterogeneity to support distinct types of V2a-like neurons also exists. In lampreys, vertical navigation using dorsal and ventral axial muscles is proposed to be driven by interconnected V2a-like neurons[15,17,61]. Paired-recordings and post-hoc fills have revealed excitatory interneurons with descending or bifurcating morphologies and also mixed or glutamatergic synaptic connections, consistent with our findings[73–75]. Similarly, tadpoles of the protovertebrate Ciona intestinalis swim vertically in the water column, but use a very rudimentary sheet of axial muscle[76]. Recent work using electron microscopy has revealed that ipsilaterally projecting last-order excitatory interneurons in their motor ganglion (MGINs) can also be distinguished based on either stronger higher order or stronger last-order connections[77]. MGINs labeled by the Chx10 homolog Vsx are also distinguished by birthdate and size[78,79]. These observations suggest that the molecular developmental programs and arrangement of distinct excitatory interneurons into synaptically stratified microcircuits predates the emergence of epaxial and hypaxial subdivisions.

Another possibility is that the conceptual framework revealed here is specific to the evolution of increased aquatic maneuverability in teleosts[80], while a true multi-layer framework evolved during the transition to land[81], perhaps by eliminating relatively weak connections (i.e., last-order type I V2a synapses and higher-order type

II V2a synapses). Determining which scenario holds true awaits similar fine-grained assessments linking neuronal identity and synaptic connectivity to function in other vertebrates.

## Methods

**Animals and interneuron targeting**. Adult zebrafish (Danio rerio) and their offspring were maintained at 28.5 °C in an in-house facility (Aquatic Habitats). Experiments were performed using 4–5-day-old wildtype, Tg[Chx10:GFP][19], Tg[Chx10:lRl-GFP][19], Tg[MNET2:GFP][82], Tg[GlyT2:GFP][33], Tg[Dbx:cre], and Tg[GlyT2:lRl-Gal4;UAS:GFP] zebrafish larvae[41]. At this stage, zebrafish larvae have fully inflated swim bladders and are free swimming, but have not yet sexually differentiated. In addition to using different combinations of stable transgenic lines to target different interneuron classes, in cases where crosses were incompatible, we used the Gal4-UAS system to transiently drive reporter constructs to achieve mosaic labeling of V2a neurons[83]. For mosaic labeling, Chx10:Gal4 was co-injected with either UAS:mcd8GFP or UAS:ptagRFP plasmids into one-cell-stage to four-cell-stage embryos using a microinjector (IM300, Narishige). DNA solutions were prepared at concentrations between 15 and 25 ng/µl. Larvae were then screened on an epifluorescent microscope (SteREO Discovery V20, Zeiss) for labeling of appropriate V2a types. All procedures conform to NIH guidelines regarding animal experimentation and were approved by Northwestern University Institutional Animal Care and Use Committee.

**Electrophysiology and imaging**. Zebrafish larvae were first immobilized using extracellular solution containing α-bungarotoxin (0.1% w/v; composition in mmol/l: 134 NaCl, 2.9 KCl, 2.1 MgCl$_2$, 10 HEPES, 10 glucose, 2.1 CaCl$_2$, adjusted to pH 7.8 with NaOH) and then transferred to a Sylgard-lined glass-bottom dish containing toxin-free extracellular solution. Larvae were pinned down through the notochord using custom sharpened tungsten pins, then the skin was carefully removed using fine forceps. For paired recordings, muscle segments overlying the spinal cord were carefully dissected away using tungsten pins and fine forceps. Following the dissection, the preparation was moved to the recording apparatus (AxioExaminer, Zeiss) equipped with a ×40/1.0 NA water immersion objective and three motorized micromanipulators (PatchStar/MicroStar, Scientifica).

For whole-cell recordings, standard wall glass capillaries were pulled to make recording pipettes with resistances between 5 and 15 MΩ, which were then back-filled with patch solution (composition in mmol/l: 130 K-gluconate, 2 MgCl$_2$, 0.2 EGTA, 10 HEPES, 4 Na$_2$ATP, adjusted to pH 7.3 with KOH) containing either Alexa Fluor 488 or 568 hydrazide (final concentration 50 µmol/l) to visualize cell morphology at the end of experiments using a cooled CCD camera (Rolera-XR, Q-Imaging). Images were captured using QCapture Suite imaging software (Q-Imaging). Electrophysiological recordings were acquired using a Multiclamp 700B amplifier, a Digidata series 1322 A digitizer, and pClamp software (Molecular Devices). Standard corrections for bridge balance and electrode capacitance were applied in current-clamp mode. Electrical signals from spinal cells were filtered at 30 kHz and digitized at 63 kHz at a gain of 10 (feedback resistor, 500 MΩ).

For paired whole-cell recordings, connectivity was assessed by delivering 5 ms step pulses at a low frequency (2 Hz) to elicit a single spike in the presynaptic cell while assessing postsynaptic responses in current clamp mode and voltage clamp mode (holding potential, −65 mV). For voltage clamp recordings, no series compensation was used. For pharmacological experiments to test the nature of connectivity, the glutamate receptor antagonist NBQX (Abcam) was dissolved in extracellular solution and delivered to the perfusate (10 µM final concentration) by a gravity-fed perfusion system. The gap junctional blocker 18-beta-glycyrrhetinic acid (Sigma-Aldrich) was first dissolved in DMSO to obtain a stock solution at 200 mM and was then used at a final concentration of 100–150 µM in extracellular solution.

For paired V2a soma-axon recordings, fish with sparse V2a neuron GFP expression following DNA injection as described above were screened under an epifluorescence dissecting scope and selected for experiments. Once a stable whole-cell recording was established at the soma, fluorescence with a high-attenuation filter was used to target the axon. A loose-patch seal was achieved by applying light suction (<15 mmHg) onto the axon using glass electrodes (4–8 MΩ resistance) filled with extracellular solution. Axonal signals were recorded in current-clamp mode.

For re-patch recordings, whole-cell recordings were first achieved between a V2a neuron and a primary motor neuron. Once the nature of the connection was tested in current-clamp mode, the morphology of the motor neuron was then captured. Afterwards, the electrode from the first recorded motor neuron was retracted and a second motor neuron was then targeted with a new electrode within the same segment while maintaining the V2a neuron recording. The same protocol was repeated to assess synaptic connectivity and confirm motor neuron morphology.

To simultaneously monitor fictive motor activity using peripheral motor nerve recordings, pipettes were fashioned from the same ones used for whole-cell recordings. However, the tip was cut to make a 20–50 µm opening and fire polished to bend the electrode to accommodate for the approach angle. To stimulate fictive swimming, a tungsten concentric bipolar electrode was lowered onto the skin using

a manual micromanipulator (YOU-3; Narishige) and a brief electrical stimulus (5–20 V; 0.2–0.4 ms) was delivered via an isolated stimulator (DS2A-Mk.II; Digitimer). Extracellular signals from the peripheral motor nerves were amplified using a differential AC amplifier (model 1700; A-M Systems) at a gain of 1000 and digitized using Digidata 1322 A with low-frequency and high-frequency cutoffs set at 300 and 5000 Hz, respectively. The recruitment pattern during fictive swimming for V2a and V0d neurons was assessed in current-clamp mode while simultaneously recording peripheral signals.

To image transgenic lines using confocal microscopy, larvae were first anesthetized in a 0.02% solution of MS-222 (Sigma-Aldrich) and then embedded in low melting point agarose (1.4% in system water) in a glass bottomed dish and covered in 10% Hank's solution with anesthetic. Fish were oriented such that the spinal cord was imaged from a lateral view. Image Z-stacks were acquired with a confocal microscope (LSM 710, Zeiss) using a ×20/1.0-NA water-immersion objective. Fluorescence images were collected concurrently with a differential interference contrast (DIC) images to allow for analysis of soma cross-sectional area and dorsoventral soma position normalized to the top and bottom aspects of spinal cord using ImageJ software.

**Quantification and statistical analysis**. All electrophysiological data were analyzed off-line using Igor Pro 6.2 (Wavemetrics) and tabulated in Microsoft Excel. The intrinsic firing pattern of V2a and V0d neurons was determined by applying 500-ms depolarizing current pulses up to a maximum of 3× threshold current. Input resistance was determined by taking the average of at least three 500-ms-long hyperpolarizing pulses (10–50 pA) within a linear range of the current–voltage relationship. Instantaneous spike frequencies were calculated from the interval of the first two spikes.

The conduction velocity of V2a cells was determined by measuring the distance between the somatic and axonal recording sites and dividing the time difference between the peak of the spike dv/dt and the peak of the axonal response. The distance between recording sites was measured using ImageJ by tracing the labeled axon in 2D from the base of the soma to the tip of the axonal electrode (mean recording distance ± standard deviation: 389.1 ± 49.6 µm, n = 11). Three distance measurements were taken and the average distance value was used for the conduction velocity calculation. The amplitude of the axonal spike was measured from peak to trough and the amplitude of the somatic spike was measured from spike threshold to peak value.

To quantify synaptic connectivity between cells we aligned presynaptic spike sweeps to the peak of the spike (10–200 sweeps per paired recording). EPSPs were smoothed (5 point boxcar averaging, 10 KHz sampling rate) and the amplitude of the response was calculated by subtracting the averaged baseline value taken 2 ms prior to presynaptic spike from the peak response value. In order to get an accurate amplitude measurement of the second glutamatergic component in mixed synaptic responses, the early electrical component was subtracted out. This was achieved by taking the average from electrical responses where the glutamatergic component was visibly absent and subtracting it from all the sweeps in that trial.

Synaptic rise times for electrical synapses (slow and fast) were calculated from 10 to 90% of the peak amplitude. Decay times were calculated by a single exponential fit between peak amplitude and baseline of the response. Connection probability was determined by counting the number of recordings with connections and dividing by the total number of recordings. Synaptic failures were determined by dividing the number of responses without a fast synaptic event by the total number of spike-triggered events.

To analyze firing reliability at different speeds of swimming, swim bursts were binned at 10 Hz and the likelihood of spiking (expressed as percent of total bursts) at any given speed bin was determined for each cell. The spike timing of V2a neurons relative to motor neuron spiking was determined only from paired recordings from which both cells were located within the same segment. Relative spike timing in reference to local motor activity was calculated by subtracting the time of the first spike in the primary motor neuron (time, 0) from the first spike in the V2a neurons. V2a spike times were binned at 0.25 ms intervals and a histogram was constructed for each neuron expressed as a percent of total spikes. These histograms were then averaged and fit with a Gaussian function.

The coupling coefficient between two electrically coupled neurons was calculated as the ratio of membrane potential deflections (ΔV2/ΔV1) in response to a 500-ms hyperpolarizing current pulse. ΔV1 is the change in membrane voltage in the cell receiving the hyperpolarizing step pulse and ΔV2 is the change in membrane voltage in the coupled cell. All data were tested for normality and the appropriate statistical analysis was performed using the StatPlus plug-in for Excel (Microsoft). Data are reported as means ± standard deviations, with corresponding tests, critical values and degrees of freedom.

**Reporting summary**. Further information on research design is available in the Nature Research Reporting Summary linked to this article.

## Data availability
All data are available from the corresponding author upon reasonable request. Data underlying Figs. 1f–h, 2c, e, f, 3c, d, 4b, d, 5d, e, 6a and 7b, f are provided as a Source Data file.

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

## Acknowledgements

We thank Elissa Szuter for fish care and Eli Cadoff for technical support. We also thank Marco Gallio, Andrew Miri, Christopher Del Negro, Keith Sillar, Joseph Fetcho, and members of the lab for feedback on numerous iterations of the paper. Financial support provided by NIH awards R01 NS067299 to D.M. and U19 NS104653 to Florian Engert, Jeff Lichtman and Haim Sompolinsky (subcontract to D.M.).

## Author contributions

Conceptualization: E.M. and D.M.; Methodology: E.M. and D.M.; Investigation: E.M., Formal analysis: E.M.; Writing: E.M. and D.M.; Visualization: E.M. and D.M.; Resources, supervision and funding acquisition: D.M.

## Additional information

**Competing interests:** The authors declare no competing interests.

