## [Peer Review File · Nature Communications]

Reviewers' comments:

Reviewer #1 (Remarks to the Author):

For decades in the locomotor field, it has been proposed that interneurons are organized in a dedicated manner either contributing to rhythm formation or pattern generation. McCrea and Ribak proposed in particular in 2008 that a clear separation of the networks involved in rhythm generation and pattern formation, with the latter only connecting to motoneurons.

In this study, Meleanou and McLean provide evidence against the strict hierarchical organization of high order and low order interneurons respectively dedicated to rhythm generation and pattern formation. They have performed an impressive amount of work (double patch clamp recordings of V2a, conduction velocity measurements, projections onto motoneurons, VOD interneurons) to support this claim.

The authors focus their analysis on spinal V2a interneurons, previously involved in speed control.

As found in the mouse, the authors show based on intrinsic properties (input resistance, rheobase, firing pattern and frequency) and morphology (functional anatomy) for a very clear distinction of two types of V2a interneurons recruited at high speed.

The authors demonstrate that 1) these two types of V2a interneurons are recruited differently relative to motoneuron spiking within the fictive fast locomotion, 2) that they are projecting differently among V2a types and onto motoneurons and inhibitory VOD interneurons.

Their data on V2a interneurons dedicated to fast swimming reveal that the connectivity pattern of interneurons involved rhythm generation and pattern formation is much more complex than previously proposed (McCrea and Ribak 2008).

The authors performed a fine analysis of connectivity based on double cell patch clamp from V2a interneurons and their target recorded in V Clamp and C clamp mode. V2a interneurons are coupled to their targets via either electrical synapses, pure glutamatergic synapses, mixed electrical and chemical synapses.

The authors provide a comprehensive view on the organization of V2a interneurons, and make a real effort to compare to the body of work performed in mammals in order to draw important conclusions.

Their work also reveals an intriguing phenomenon: between V2a interneurons, a slow current (long rise time and long decay time) can be observed - most likely corresponding to indirect coupling via reticulospinal axons, as a similar slow current can be observed between pairs of motoneurons that are 4 segments away, and therefore which cannot be directly connected. This is very interesting as it may provide an alternative explanation for some of the previous observations done in the field.

Major comments

The data presented here speak for itself and I only have comments to go slightly deeper in the understanding of general principles that could guide connectivity

1. Can the authors comment on the level of electrical coupling versus synaptic connectivity and the morphological overlap of presynaptic and postsynaptic targets ?

2. In the same line, are GAP junctions mainly occurring between soma and dendrites of V2a neurons? Is there evidence for a preferred compartment of the neuron being forming GAP junctions ? IHC for connexin could be informative in this regard.

Writing

- It would be good to be more precise and specific regarding the class of interneurons targeted in the abstract.
- As the results obtained on V2a connectivity are complex, the first and last schematic should be more extensively detailed regarding the model proposed and the conclusions that can be made from the present study.

Figures

- Electrophysiological data should be expanded/larger panels so that readers can fully appreciate the amplitude of the synaptic or coupling effect in the traces (for example in Fig 6 but not only).
- It would be nice to keep the scale bars with values in the panel (if possible with Nature Comm policy).

Reviewer #2 (Remarks to the Author):

In this manuscript, Menelaou and McLean performed detailed analyses of spinal V2a neurons in larval zebrafish. The authors focused on V2a neurons that are exclusively engaged in high frequencies of swimming (dorsally-located V2a neurons), and carried out an amazing amount of work, which included a couple of hundreds of paired patch recordings. The purpose of the study was to see how spinal interneurons are synaptically interconnected to explain the basis of timing and amplitude control during locomotion (fast swimming in the case of this study). The authors first showed that V2a neurons were classified into two types, type I and type II, as in mice. The authors then revealed that distinct V2a neurons exhibit complementary patterns of higher-order and last-order connectivity, with stronger connections from type I neurons to other excitatory and inhibitory neurons for frequency and phase control, and stronger connections from type II neurons to motor neurons for amplitude control. Thus, timing and amplitude are controlled by interconnected sets of V2a neurons with differences in the weights and probabilities of higher-order and last-order connections during fast swimming in larval zebrafish.

In higher vertebrates, it is generally believed that higher-order interneurons lacking direct motor neuron connections control the timing of movements, while last-order interneurons with direct motor neuron connections control movement amplitude. The findings of this manuscript present a new view: like zebrafish, timing and amplitude in mammalian locomotion may be controlled by interconnected sets of interneurons with differences in the weights and probabilities of higher-order and last-order connections. In this sense, this paper will have a wide-ranging impact on the research field of spinal locomotor circuits, and hence, I strongly recommend publication of this manuscript in Nature Communications.

Minor comments:

1) Line 38 (and many other places): The authors use *Alx* as the gene name for the marker of V2a neurons. However, this nomenclature had been only used in very early literature. For the past 10 years, *Chx10* or *Vsx2* has been used in most literature. Indeed, in the publication of the same authors (Menelaou et al., 2013, *Journal of Comparative Neurology*), they used the term, *Chx10*. It is not clear why the authors returned to *Alx*, but I feel that *Chx10* would be more appropriate because this nomenclature is the most commonly used one in the research field of spinal

locomotor circuits.

2) Line 173: The authors used Tg[Glyt2:IRI-GFP x Dbx1:Cre] or Tg[Glyt2:GFP] for the targeted recordings of V0d neurons. While the former only marks V0d neurons, the latter is not specific to V0d neurons. Thus, there is a possibility that commissural inhibitory neurons other than V0d neurons might be included in the data sets. This point needs to be mentioned.

3) Line 187: "Fig. 53" should be "Fig. 5e".

4) Line 356: "Recent work using electron microscopy has revealed that last-order excitatory interneurons in their motor ganglion can also be distinguished by stronger higher order versus stronger last-order connections".

I could not find the corresponding data in Ryan et al. (2017). I may have missed it, but if the authors consider the ddN neuron as the stronger higher order interneuron, this interpretation may not be appropriate. This is because the ddN neuron is regarded as a homolog of the Mauthner cell, which is completely different from the other last-order neurons.

5) Figures 3c and 3d: It is not very clear whether the data shown in the panels includes only rostral-to-caudal connections or both rostral-to-caudal and caudal-to-rostral connections. This should be described in the figure legend.

6) Line 718: "lower-order" should be "last-order".

Reviewer #3 (Remarks to the Author):

The manuscript by Menelaou and McLean examines in detail the connectivity of V2a neurons, which are the primary class of ipsilaterally-projecting excitatory neurons in the spinal cord. The authors describe in great detail the nature of electrical and chemical synaptic connectivity between two types of V2a neurons, and between these two V2a subtypes and fast axial motor neurons, as well as inhibitory commissural neurons that control left-right alternation. Important nuanced differences in firing properties, synaptic strength, electrical coupling, axon conduction speeds are noted. This is a carefully performed and detailed study that provides valuable information about two subpopulations of V2a neurons and their connections within the spinal cord of larval zebrafish.

While one can have great confidence in the state-of-the-art technical approaches that have been used and the experimental measurements that have been made, the implications of their findings for motor circuit organization are less clear. Much of the discussion is speculative, with arguments being built on the premise of the 2 layered model proposed by Rybak, McCrea and colleagues. This separates the upstream spinal interneurons into higher order (rhythm generating) and lower order (pattern generating) groupings, for which there is limited evidence. Thus, some care needs to be taken when classifying V2a type 1 neurons as higher order and type II neurons as lower order. The authors attempt to put forward a model that encompasses their findings, however the description of this model and the predictions it makes are rather limited, nor are there any experiments that directly test this model. At a minimum, the authors need to provide a better description of their model and how it compares/contrasts with previously proposed locomotor circuit models including the Rybak model. References to higher order and lower order neurons and the associated implications that are present throughout the manuscript also need to be tempered and placed in a fuller context.

We are extremely grateful to the reviewers to their thoughtful comments and have now had the opportunity to revise the manuscript accordingly. Please find below a point-by-point response. While none of the reviewers expressed any problems with the data, there were some concerns about how we framed the work and discussed the implications. We have made substantial changes to the title, abstract, introduction, discussion and figures to address these issues. As a result, the manuscript is greatly strengthened, so thank you again for your time and effort.

Reviewer #1

Major comments

The data presented here speak for itself and I only have comments to go slightly deeper in the understanding of general principles that could guide connectivity

1. Can the authors comment on the level of electrical coupling versus synaptic connectivity and the morphological overlap of presynaptic and postsynaptic targets?

We cannot rule out electrical coupling between neurons in the same segment arises from dendritic overlap. Therefore, to address the second issue and to also provide a more accurate comparison to the motor neuron data, we have restricted our analysis to V2a paired recordings performed 2-4 segments apart, where only axonal interactions from presynaptic neurons are possible. It can be seen from the new Fig. 6e that coupling of a similar magnitude is still observed as witnessed between motor neurons. To address the first issue, we have performed analysis of coupling coefficients in the caudo-rostral direction related to the type of synaptic connection involved, again only between neurons separated by 2-4 segments. This now reveals that coupling is present of equal magnitude regardless of the nature of V2a connection (slow, electric, glutamatergic or mixed). We think this further bolsters the argument that coupling is likely an independent phenomenon unrelated to direct synaptic connections and more likely reflects common inputs. In addition to a new figure panel and associated text, the main text has been modified as follows (new L222):

“Given the distinctive nature of slow responses and the preponderance of electrical synaptic connections, we were concerned that slow responses may in fact represent indirect electrical continuity, either via common target neurons⁴² and/or a shared source of reticulospinal input⁴³. To test the latter possibility more directly, we measured electrical coupling coefficients between primary motor neurons separated by 4 body segments, where there is no chance of a direct physical contact (Fig. 6d). In support of indirect electrical continuity, we observed bidirectional propagation of hyperpolarizing current steps between distal motor neurons (Fig. 6e, f). When we performed a similar analysis between different types of V2a neurons separated by 2-4 body segments, where physical contact via presynaptic axons is the only possibility, a similar level of coupling was observed (Fig. 6e, g, h). Remarkably, however, when assessing V2a neurons and all of their post-synaptic targets, electrical coupling was observed in the caudo-rostral direction regardless of the mode of synaptic transmission, whether slow, electrical, glutamatergic or mixed (Fig. 6i). Collectively, these data suggest that electrical coupling recorded between distal neurons does not report the existence of fast interconnections and that slow responses are more likely to reflect indirect electrical interactions.”

2. In the same line, are GAP junctions mainly occurring between soma and dendrites of V2a neurons? Is there evidence for a preferred compartment of the neuron being forming GAP junctions ? IHC for connexin could be informative in this regard.

Because we have now focused on recordings between neurons separated by 2-4 body segments coupling between the soma and dendrites of neurons is unlikely, since dendrites do not extend that far at this stage. Our previous anatomical work suggests that fast V2a connections are likely perisomatic based on axon trajectories. We now add a statement to the following sentence to acknowledge this (new L42):

“This drive could originate from two sources, V2a neurons with descending projections within spinal cord and V2a neurons with descending and supraspinal projections, both of which appear to make perisomatic last-order connections²².”

A more comprehensive analysis of the subcellular distribution of gap junctions in the different types of neurons studied here in larvae is certainly something we plan to follow up on. However, since it is not the primary goal of what we're focused on here, we hope mentioning the likely perisomatic origin of V2a axonal contacts to motor neurons is sufficient to address this concern.

Writing

- It would be good to be more precise and specific regarding the class of interneurons targeted in the abstract.

We have added V2a and V0d nomenclature to the abstract to identify the types of interneurons investigated here. We have also added V2a neurons to the title.

- As the results obtained on V2a connectivity are complex, the first and last schematic should be more extensively detailed regarding the model proposed and the conclusions that can be made from the present study.

We have modified the first and last schematics to provide more detail about the cell types involved. This includes a new Fig. 7, which allows for comparisons across models to be more easily compared.

Figures

- Electrophysiological data should be expanded/larger panels so that readers can fully appreciate the amplitude of the synaptic or coupling effect in the traces (for example in Fig 6 but not only).

We have expanded the electrical coupling traces in Figure 6 and also increased the size of all the figures so that the data is more easily appreciated. We would be happy for the figures to be displayed full page width if necessary.

- It would be nice to keep the scale bars with values in the panel (if possible with Nature Comm policy).

We have added values to all scale bars. Although this is not strictly their policy, we do think it is more helpful to have them close by, so we hope an exception can be made to keep them there.

Reviewer #2

Minor comments:

1) Line 38 (and many other places): The authors use *Alx* as the gene name for the marker of V2a neurons. However, this nomenclature had been only used in very early literature. For the past 10 years, *Chx10* or *Vsx2* has been used in most literature. Indeed, in the publication of the same authors (Menelaou et al., 2013, Journal of Comparative Neurology), they used the term, *Chx10*. It is not clear why the authors returned to *Alx*, but I feel that *Chx10* would be more appropriate because this nomenclature is the most commonly used one in the research field of spinal locomotor circuits.

*We have replaced all references in the main text to *Alx* with *Chx10*.*

2) Line 173: The authors used *Tg[Glyt2:IRI-GFP x Dbx1:Cre]* or *Tg[Glyt2:GFP]* for the targeted recordings of V0d neurons. While the former only marks V0d neurons, the latter is not specific to V0d neurons. Thus, there is a possibility that commissural inhibitory neurons other than V0d neurons might be included in the data sets. This point needs to be mentioned.

We have added a sentence acknowledging the possibility that CoBL interneurons can also derive from another, albeit more dorsally-distributed, source, as follows (new L180):

*"By focusing on ventral-most cells in the *Tg[GlyT2:GFP]* line, this also reduced the possibility of recording from CoBL interneurons derived from more dorsal progenitor domains⁴¹. Post-hoc fills using either approach revealed neurons with indistinguishable morphologies and recruitment patterns, namely a spherical soma with short dendritic processes from the ipsilateral axon and collaterals that project dorsally from the commissural bifurcating axon (Fig. 5b)."*

3) Line 187: "Fig. 53" should be "Fig. 5e".

Changed.

4) Line 356: "Recent work using electron microscopy has revealed that last-order excitatory interneurons in their motor ganglion can also be distinguished by stronger higher order versus stronger last-order connections". I could not find the corresponding data in Ryan et al. (2017). I may have missed it, but if the authors consider the ddN neuron as the stronger higher order interneuron, this interpretation may not be appropriate. This is because the ddN neuron is regarded as a homolog of the Mauthner cell, which is completely different from the other last-order neurons.

This was an Endnote error on our part and the wrong reference was inserted here, thanks for pointing this out. We now include the correct reference and also name the interneurons to avoid any confusion, as follows (new L361):

"Recent work using electron microscopy has revealed that ipsilaterally-projecting last-order excitatory interneurons in their motor ganglion (MGINs) can also be distinguished based on either stronger higher order or stronger last-order connections⁷⁶. MGINs labeled by the Chx10 homolog Vsx are also distinguished by birthdate and size^{77, 78}."

The paper we meant to cite describes MGINs, which are ipsilaterally projecting and include neurons labeled by Vsx. We also provide a review which links MGINs to Vsx expression (new #78).

5) Figures 3c and 3d: It is not very clear whether the data shown in the panels includes only rostral-to-caudal connections or both rostral-to-caudal and caudal-to-rostral connections. This should be described in the figure legend.

We have modified the legend to include 'descending' so it is clear that it is in the rostral to caudal direction in both cases.

6) Line 718: "lower-order" should be "last-order".

Changed.

Reviewer #3 (Remarks to the Author):

The manuscript by Menelaou and McLean examines in detail the connectivity of V2a neurons, which are the primary class of ipsilaterally-projecting excitatory neurons in the spinal cord. The authors describe in great detail the nature of electrical and chemical synaptic connectivity between two types of V2a neurons, and between these two V2a subtypes and fast axial motor neurons, as well as inhibitory commissural neurons that control left-right alternation. Important nuanced differences in firing properties, synaptic strength, electrical coupling, axon conduction speeds are noted. This is a carefully performed and detailed study that provides valuable information about two subpopulations of V2a neurons and their connections within the spinal cord of larval zebrafish.

While one can have great confidence in the state-of-the-art technical approaches that have been used and the experimental measurements that have been made, the implications of their findings for motor circuit organization are less clear. Much of the discussion is speculative, with arguments being built on the premise of the 2 layered model proposed by Rybak, McCrea and colleagues. This separates the upstream spinal interneurons into higher order (rhythm generating) and lower order (pattern generating) groupings, for which there is limited evidence. Thus, some care needs to be taken when classifying V2a type 1 neurons as higher order and type II neurons as lower order. The authors attempt to put forward a model that encompasses their findings, however the description of this model and the predictions it makes are rather limited, nor are there any experiments that directly test this model. At a minimum, the authors need to provide a better description of their model and how it compares/contrasts with previously proposed locomotor circuit models including the Rybak model. References to higher order and lower order neurons and the associated implications that are present throughout the manuscript also need to be tempered and placed in a fuller context.

The reviewer is supportive in their assessment of the data, but has three major concerns (#1-3) and two suggestions (#5,6), which we have addressed by substantial modifications to the title, abstract, introduction, discussion and figures. We summarize the edits below, but for a full accounting please refer to the manuscript file where all changes are marked in red:

- 1) *The implications of our findings for motor circuit organization are unclear.*

In response to Reviewer #1, we have now modified Figure 1 to be more explicit about the predictions of two leading models and added a new Figure (Fig. 7) which allows for our findings to be more easily compared to these predictions. To further address this concern, the title has been changed to more clearly state the discovery. The abstract has also been changed to more succinctly define the question that is still open in the field and the introduction now sets up the problem with respect to differences in the predictions of two prevailing models (multi-layer and single layer) and conflicting experimental evidence. We have also included more references for the reader to turn with other sources that highlight the lack of consensus regarding whether the same interneurons are responsible for timing versus amplitude control (new #10, 11, 14). We appreciate the opportunity to more clearly state for readers that this is a fundamental unsolved issue in the field, which our data now help explain more easily.

- 2) *Much of the discussion is speculative.*

We have re-written the Discussion to focus on comparing and contrasting the model with others and the functional implications.

- 3) *The description of this model and the predictions it makes are rather limited, nor are there any experiments to directly test this model.*

Our goal here was to test the predictions made by previous models in the most direct way possible, by performing paired recordings. Instead of supporting one or the other, we find that aspects of both are in play but in a way not completely predicted by either model. Since the power of any model lies in its explanatory power, the discussion now focuses on how our model better reconciles previous experimental evidence, in addition to detailing the functional significance for zebrafish locomotion and spinal motor control more broadly. We have also added more predictions, which can now be tested on the basis of our work here.

- 4) *Provide a better description of their model and how it compares and contrasts with previous models.*

We have modified the introduction and discussion, in addition to providing more detailed circuit diagrams in Figure 1 and 7.

- 5) *References to higher order and lower order neurons and associated implications need to be tempered and placed in fuller context.*

We have ensured that 'higher-order' and 'last-order' are now exclusively in reference to the connections, defined by whether they are to interneurons (higher-order) or motor neurons (last-order). Any references to higher order or last order interneurons are raised in the context of predictions arising from multi-layer versus single layer models. We no longer refer to 'rhythm-generating' or 'pattern-forming' to avoid the associated implications. As mentioned above, we have also provided more references for context (new # 10, 11, 14).

REVIEWERS' COMMENTS:

Reviewer #1 (Remarks to the Author):

The authors very satisfactorily answered my questions and I very much enjoyed reading the revised discussion and additional points brought to the Results section.

The analysis of electrical coupling among distant neurons will be very valuable to the field, and brings a lot to our understanding of unpublished observations as well as recent publications.

Good job !

Reviewer #2 (Remarks to the Author):

The authors satisfactory addressed the points that I raised.

Reviewer #3 (Remarks to the Author):

In probing the nature of V2a neuron connectivity Menelaou and McLean provide important insights into the structure of the spinal locomotor circuitry. The authors now provide a much clearer rationale for their experimental findings, particularly in regards to prior models for the locomotor CPG. The discussion is extremely thoughtful and provoking and will serve as a conceptual guide for further studies on spinal motor circuit organization in aquatic and limbed vertebrates, with important implications for locomotion and other spinally driven motor behaviors. In sum, this is an outstanding study that makes a valuable contribution to the field.